

# Occurrence patterns of sympatric forest wallabies: assessing the influence of structural habitat attributes on the coexistence of *Thylogale thetis* and *T. stigmatica*

Lucy E.V. Smith[1], Nigel R. Andrew[2,3] and Karl Vernes[1,2]

[1] Ecosystem Management, University of New England, Armidale, New South Wales, Australia
[2] Natural History Museum, University of New England, Armidale, New South Wales, Australia
[3] Faculty of Science and Engineering, Southern Cross University, Lismore, New South Wales, Australia

Corresponding author
Karl Vernes, kvernes@une.edu.au

## ABSTRACT

**Background**. We studied the occurrence of two sympatric wallabies, the red-necked pademelon (*Thylogale thetis*) and the red-legged pademelon (*T. stigmatica*) in north-eastern New South Wales, Australia in relation to structural habitat attributes. At our study site, both species inhabit closed forest environments and have overlapping distributions, but *T. thetis* leaves the forest at night to graze adjacent grassy forest edges whereas *T. stigmatica* remains within the forest and browses forest vegetation. The objectives of the study were to investigate how structural attributes of two forest types, wet sclerophyll forest and rainforest, relate to the fine-scale occurrence of these two wallaby species within the forested environment.

**Methods**. We gathered occurrence data from 48 camera trap stations divided equally between rainforest and wet sclerophyll forest. At each camera point, we also measured a range of structural habitat attributes to determine habitat affiliations for the two *Thylogale* species. Principal component analyses were used to describe major trends in habitat, and generalised linear models were used to describe the efficacy of the variables in predicting habitat occurrence of each species.

**Results**. The number of occurrences of *Thylogale thetis* was significantly greater than occurrences of *T. stigmatica*, which was driven by significantly greater occurrences of *T. thetis* in wet sclerophyll forest. There was both spatial and temporal partitioning between the two species; there was a significant difference in the occurrences of the two species at individual cameras and *T. stigmatica* had a different activity schedule than *T. thetis* in wet sclerophyll forest, where the latter reached its greatest rate of occurrence. At a finer (camera station) scale, occurrences of *T. thetis* increased with proximity to roads and grassy edges and at sites that were less rocky and less steep. *T. stigmatica* occurrence increased in the presence of rainforest elements like vines, palms and ferns, more ground-level cover and tree-fall gaps and at sites with fewer emergent eucalypts.

**Conclusion**. Our findings have implications for managing these pademelons and their habitats. *T. thetis* is a common species that was encountered more often than *T. stigmatica*, and it responded positively to human disturbance like roadsides and grassy edges, presumably because these areas provided good grazing opportunities. By comparison, *T. stigmatica* is a threatened species, and it responded to natural

disturbance like tree-fall gaps where lateral cover was greater, and where rainforest food plants may be more abundant. Our results suggest, therefore, that conservation of the threatened *T. stigmatica* requires the preservation of intact rainforest.

## INTRODUCTION

Ecological differences in species that allow for niche partitioning, and therefore coexistence, are manifested in three main ways. Species may differ in the resources on which they specialise (resource partitioning), conversely, they may partition their activity in time (temporal partitioning) and/or they may differ their activity in space (spatial partitioning) (*Amarasekare, 2003*). Habitat heterogeneity plays a key role in species richness and the ability of species with similar resource needs to co-occur, because an increase in the variety and structural complexity of available habitat types and the resources they support increases available niche space and allows more species to coexist (*Stein, Gerstner & Kreft, 2014*; *Tews et al., 2004*).

In forested ecosystems, plant communities largely determine the physical structure of the environment and therefore influence the structure of animal communities, species richness and the coexistence of species. Vegetation may drive the fine-scale spatial distribution of sympatric species by dictating the availability of resources at multiple scales (*Kubiak, Galiano & de Freitas, 2015*), both spatial and temporal. Drawing on markedly different studies in the same forest type highlights the importance of vegetation; in Brazil's Atlantic Forest, bat diet was best explained by landscape composition, particularly vegetation density (*Oelbaum et al., 2022*), while in the same forest, the diversity of ants was explained by a combination of shrub leaf density and tree circumference (*Sampaio et al., 2023*). Among macropods, *Le Mar & Mcarthur (2005)* showed that sympatric wallabies in Tasmania had similar food requirements and foraged in the same habitats by night, but that their selection of daytime refuges differed markedly, probably due to their contrasting predator avoidance strategies.

Competitive interactions between species can also play an important role in shaping mammalian community assemblages. Competition may manifest in a number of ways, including behavioural changes, shifts in diet, or differential use of preferred habitat in space and time to avoid competition and other unfavourable encounters (*Karanth et al., 2017*). Historical competition between sympatric species may shape their current spatial distribution, resource use, and even phenotypic traits. For example, competitive interactions between sympatric bat species may lead to character displacement, where species exhibit phenotypic changes that reduce competition along one or more resource axes (*Shi et al., 2018*). Equally plausible, however, is that differences in resource use relate to evolutionary pressures developed in isolation before the two species were brought together.

Regardless, populations of closely related species are considered to be sympatric even if they are ecologically distinct, provided a high proportion of each population encounters individuals of the other along adjacent or shared ecotones (*Mallet et al., 2009*).

Anthropogenic fragmentation can also play a role in shaping competitive interactions because fragmentation can alter habitat structure through edge effects and reduction of the overall amount of original habitat, but it can also create new opportunities for species that are pre-adapted to exploit the matrix of modified habitats in a fragmented landscape (*Laurance, 1994*; *Laurance, 1997*). Habitat fragmentation can therefore change the way closely related species interact (*Valiente-Banuet et al., 2015*). In Australian rainforests, fragmentation has a strong impact on mammalian assemblages (*Laurance, 1997*) and these changes might not be evident until many decades post-disturbance (*Laurance, Laurance & Hilbert, 2008*).

In Australia, two species of rainforest-dwelling wallaby occur in eastern Australia. The red-necked pademelon *(Thylogale thetis)* inhabits rainforest as well as other forest vegetation types with a dense understorey in eastern Australia's subtropics and is most common at forest edges adjacent to pasture (*Jarman & Phillips, 1989*) where it grazes at night on pasture edge close to the forest. The red-legged pademelon (*T. stigmatica*) has a wider distribution than *T. thetis*, occurring from the extreme northern tropics at Cape York in northern Queensland to the mid north coast of New South Wales (*Johnson & Vernes, 2008*). In the northern part of its range where it occurs in the absence of *T. thetis*, *T. stigmatica* spatio-temporally partitions its range, spending diurnal hours resting and browsing within the forest interior and nocturnal hours grazing at the forest-pasture boundaries (*Vernes, Marsh & Winter, 1995*). When sympatric with *T. thetis*, *T. stigmatica* consumes only forest browse (*Calaby, 1966*; *Jarman & Phillips, 1989*; *Vernes et al., 2006*), and in northeastern New South Wales, our recent work has shown that they remain in the forest interior and avoid open grassy areas at the forest edge that are grazed by *T. thetis* (*Smith, Andrew & Vernes, 2022*). When sympatric, relative population densities can vary; *Johnson (1977)* reported *T. stigmatica* to be less abundant than *T. thetis* at an upland site at Dorrigo NSW, but *McHugh et al. (2019)* found the opposite to be the case in the North Coast Bioregion in far north-eastern New South Wales. *T. stigmatica* also appears to be less abundant in the south of its range (when in sympatry with *T. thetis*), compared to when it occurs as the sole pademelon species in the north of its range (*Vernes, Elliott & Elliott, 2022*). Diet and habitat usage by the different species and sub-species of pademelons in eastern Australia is also borne out in studies of dental morphology; while both *T. thetis* and *T. stigmatica* have a dental morphology suited to browsing (*Sanson, 1989*), differences in cranial morphology point towards *T. thetis* incorporating more grass in the diet than *T. stigmatica* generally, but for the northern sub-species of *T. stigmatica* to graze more than the southern sub-species (*Mitchell et al., 2018*).

In a recent study, we showed that *T. thetis* and *T. stigmatica* demonstrated strong spatiotemporal niche partitioning (*Smith, Andrew & Vernes, 2022*). The objectives of the current study were to investigate the broad-scale patterns of occurrence of these species in wet sclerophyll forest and rainforest, how the structural attributes of these forest types
relate to their finer-scale occurrence, and whether these patterns are suggestive of habitat partitioning by these sympatric species.

## MATERIALS & METHODS

### Study animals and study site

Two forest-dwelling pademelons (*T. thetis* and *T. stigmatica*) were the focal species for this study. Both are medium-sized macropods (*T. thetis*: males 2.5–9.1 kg, females 1.8–4.3 kg; *T. stigmatica*: males 3.7–6.8 kg, females 2.5–4.2 kg) that require rainforest or other closed forest vegetation for shelter and diurnal browsing (*Eldridge & Coulson, 2015*; *Johnson & Vernes, 2008*). Pademelons occupy small, stable home ranges within which they make daily habitual movements between shelter and feeding sites (*Johnson, 1980*; *Vernes, Marsh & Winter, 1995*). Both species breed year-round (*Johnson, 1977*; *Johnson & Vernes, 1994*), and neither diel activity patterns (*Smith, Andrew & Vernes, 2022*) or diet (*Johnson, 1977*; *Vernes, 1995*) are influenced substantially by season. Although similar in overall appearance, these species are easily distinguished by their colour patterns when white-flash photography is used, as was done exclusively in this study. *T. thetis* has an unmistakable rufous-coloured neck and shoulders (without rufous-coloured body and legs) and a faint white cheek stripe, whereas *T. stigmatica* has conspicuous rufous face, flanks and hind legs (and no rufous-coloured neck and shoulders), and a prominent white cheek stripe.

The study occurred in the Mount Hyland region of northeastern NSW, Australia, on the eastern slopes of the Great Dividing Range (study site centre: −30.165403°, 152.470407°; elevation range: 900–1,040 m). The area has a mild climate, with a mean maximum temperature of 20 °C and a mean minimum temperature of 10 °C (*Australian Bureau of Meteorology, 2018*). The study area of approximately 400 ha spanned private land, state forest and nature reserve (Fig. 1). Three major vegetation types occurred at the site: Northern Warm Temperate Rainforest (a wet closed forest of non-sclerophyllous tree species, with an open shrub layer, and some epiphytes and lianas; hereafter 'rainforest'), Northern Hinterland Wet Sclerophyll Forest (a tall, open eucalypt forest with an open shrubby understorey; hereafter 'wet sclerophyll forest') and an anthropogenic grassy area clear of trees or shrubs (Fig. 1). At the centre of the study area, a private parcel of land called 'Motherland' (now part of Hyland Nature Reserve) comprised native forest vegetation situated around a grassy clearing; forest on the southern side of the clearing was predominantly wet sclerophyll forest with a rainforest understorey, while rainforest dominated the northern side of the clearing. This privately-owned forested area was continuous with the larger surrounding state forests and nature reserve comprising a mix of wet sclerophyll forest and rainforest (Fig. 1). At the time the research was undertaken, the study area was privately owned and not accessible to people other than those directly involved in the research.

Ethics approval for this research was obtained from the University of New England Animal Ethics Committee (Approval No. AEC-1708), and scientific licences for this research were issued by the New South Wales (NSW) Office of Environment and Heritage (Permit Nos. SL101721 and SL101837). While ethical approval specific to human subjects
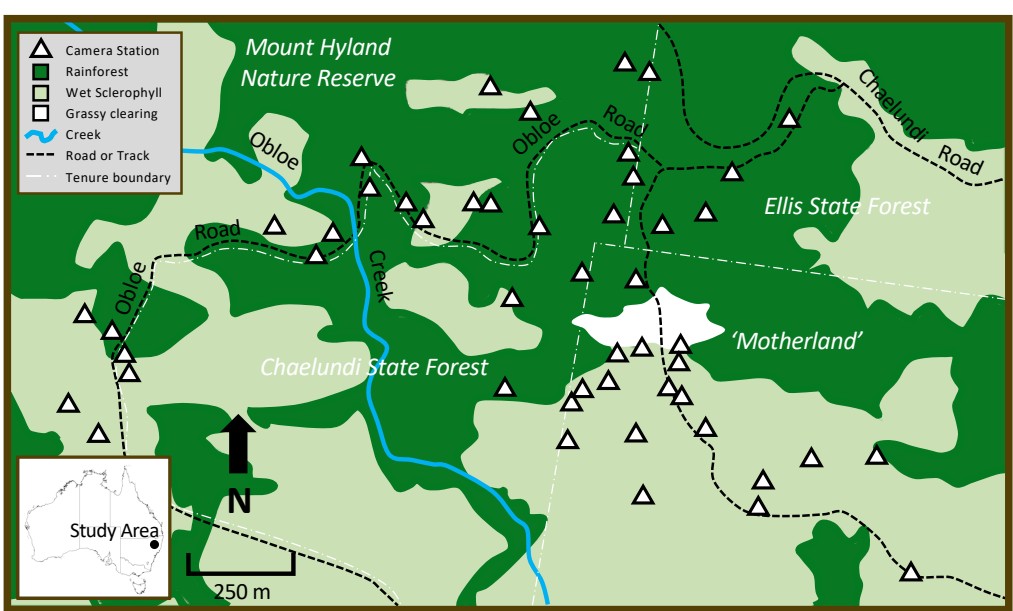

**Figure 1** **The study area in northeastern New South Wales, showing patterns in vegetation type across different land tenures, major site features, and camera trap locations.** Dark green = rainforest, light green = wet sclerophyll forest, white = grassy clearing. White triangles show position of the 48 cameras that were deployed in rainforest and wet sclerophyll forest.

being caught incidentally on camera is not required in NSW, our standard procedure is to immediately delete any images of human subjects. However, as we set camera traps off-trail within a study site restricted by locked gates, no human subjects other than those directly associated with the project were captured.

## Camera trap placement

Camera trapping methods employed were as previously described by *Smith, Andrew & Vernes (2022)*. We set forty-eight Scoutguard and UOVision white flash cameras in total: 24 were in wet sclerophyll with a rainforest understorey and 24 were in rainforest (Fig. 1). Local fire trails allowed access to the forest habitat; these tracks were narrow and maintained an overstorey that limited the growth of non-forest vegetation. Random placement of cameras was achieved by partitioning access trails throughout the site into 100 m segments. At the end-point of each segment, we generated a random compass bearing (between 0–360°) and a corresponding random distance of between 0–300 m that directed us to where the camera would be placed. Cameras were positioned on the nearest tree to the randomly-allocated point, approximately 0.5 m above ground level and facing south. Baits were used with the purpose of prolonging the time an animal spent in front of a camera; these consisted of a porous PVC canister containing cotton wool soaked with truffle oil placed about 2 m in front of each camera. Baits were inaccessible to animals. Approximately 80% of the LED flash bulbs on each camera were covered with adhesive tape to prevent photos taken in low-light or at night being 'washed out' by excessive illumination of animal subjects at close range. Vegetation in the zone between camera and
bait was pruned judiciously to avoid false triggers. Camera batteries, SD cards and baits were replaced every 8–12 weeks during a 425-day deployment from 20 January 2017 to 21 March 2018.

## Independence of camera trap images

To ensure temporal independence of records of animals at a camera trap, we took a conservative approach by considering all of the photos taken of each species during the course of a 24-hour period as one independent record of the animal at that camera. We then corrected these records for uneven numbers of camera trap nights (due to the occasional camera that failed prior to the periodic battery replacements and downloading of data) to yield independent records per 100 camera trap days. In all subsequent analyses, therefore, with the exception of our representations of temporal activity (see below), 'occurrence' indicates the number of independent records (*i.e.,* days) per 100 days of either *T. thetis* or *T. stigmatica* over the 425-day study. The only exception to this was in the calculation of temporal activity patterns; because an understanding of temporal activity patterns requires knowing all times of the 24-hour cycle an animal is active, we considered an 'independent temporal event' to be any photo of the same species at a camera separated temporally by a time gap of more than 30 min.

## Habitat variables

Habitat variables were assessed at each camera point within a 5 m × 5 m plot centred on the camera trap (Table 1). The slope of each site was measured using a clinometer. The combined forest canopy and sub-canopy cover was measured using a 'Model C' concave spherical crown densiometer (Forest Densiometers, Rapid City, South Dakota) that estimated cover based on how many of the 24 cells on the densiometer were obscured by vegetation. Four readings were taken (one in each cardinal direction from the plot centre) and a mean value calculated. Sub-canopy foliage cover was measured using an ocular tube at random points around the perimeter of the 5 m × 5 m plot, with random whole numbers between 1 and 10 used to determine the number of steps between each of the 10 measurement points. At each point, the sub-canopy was viewed through the tube's cross-hairs and scored as either '1' (cross-hairs intersecting vegetation) or '0' (cross-hairs intersecting open sky). Lateral density was estimated using a 1 × 1-m white sheet positioned on the perimeter of the plot at each of the four cardinal directions. For each measurement, an observer would stand with their back to the centre reference tree while the white grid sheet was held vertically with one edge in contact with the ground. Percentage cover was calculated by counting the number of squares obscured by vegetation. Leaf litter was measured on a 0–3 ranked scoring system according to depth. Vines, palms and ferns were measured on a 0–3 ranked scoring system according to density per square metre. The number of woody stems in two size classes were estimated according to 0–3 ranked scoring system. Trees were classed according to their diameter at breast height (DBH) then scored according to density. Rockiness of soil was also measured using a 0–3 ranked scoring system. Number of eucalypt emergents and any tree-fall gaps were counted within each plot. Distance to nearest road, forest edge and major water source (permanent creek)
**Table 1** **Vegetation and landform attributes assessed at each camera site in rainforest and wet sclerophyll forest at Mt Hyland, NSW.**

| Measurement | Unit or score | Description |
| --- | --- | --- |
| Leaf litter depth | 0–3 | 0: absent, 1: <5 cm, 2: 5–10 cm; 3: >10 cm |
| Vines, Palms, Ferns | 0–3 | 0: absent, 1: <3 per m$^2$, 2: 3–5 per m$^2$; 3: >5 per m$^2$ (for each) |
| Rockiness of soil | 0–3 | 0: absent, 1: <3 per m$^2$, 2: 3–5 per m$^2$, 3: >5 per m$^2$ |
| Fallen Timber | 0–4 | see *Maser et al. (1979)* |
| Ground Cover | 0–4 | 0: absent, 1: 1–25%, 2: 26–50%, 3: 51–75%; 4: >75% |
| Lateral cover | % | % of 1 × 1 m white grid obscured by 0–1 m high vegetation at a distance of 5 m |
| Canopy cover | % | Estimated using concave spherical crown densitometer |
| Sub-canopy cover | % | calculated from 10 presence/absence random measurements using ocular tube |
| Small stem density (>10 cm dbh) | 0–3 | 0: absent, 1: <3, 2: 3–5; 3: >5 |
| Medium stem density (10–30 cm dbh) | 0–3 | As for small stem density |
| Slope | % | Evaluated using clinometer |
| Tree fall gaps | No. of gaps | in entire plot |
| Eucalypt emergent | No. of emergent | in entire plot |
| Distance to Nearest Road | m | Linear distance |
| Distance to edge | m | Linear distance |

were also measured using a map and estimated to the nearest metre. The degree of decay of large logs and other fallen timber was scored according to a five-point scale outlined by *Maser et al. (1979)* who used bark characteristics, presence or absence of twigs, wood texture, log shape, wood colour and portion of log on the ground to estimate the degree of decomposition. No disturbances (*e.g.*, fire, flood, domestic animal grazing) occurred in the study area during the study, and all measured variables were expected to have remained constant at any one site over the duration of the work.

## Statistical analysis

All analyses were undertaken using R (*R Core Team, 2020*). A two-way ANOVA (using the package 'stats') was used to compare the occurrences of each species at cameras from the two forest sites. For this analysis, we only included those cameras that had detected at least one or the species during the study. Data were log transformed and residuals were analysed using a Shapiro–Wilk Normality Test to confirm assumptions of normality. Principal Component Analysis (PCA) was undertaken using the package 'stats' and visualized using the package 'ggbiplot'. PCA examines quantitative associations between a group of variables, summarizing them parsimoniously into fewer variables, or 'components'. We summarised the 17 habitat variables into components that described major trends in

habitat. PCA loading scores, determined for each camera location, were used to establish the impact of individual variables on each principal component. Generalised linear modelling was then used to determine the efficacy of each component in predicting the fine-scale habitat occurrences of each *Thylogale* species, the latter expressed as the total number of independent detections made of *T. thetis* and *T. stigmatica* at each camera trap. The partitioning of occurrences of the two pademelon species in space were tested using a paired *t*-test with species occurrence data paired by camera location. Temporal activity patterns were visualised by creating polar plots of independent temporal events of pademelons on camera throughout the 24-hour cycle using the package 'ggplot2'. Times were binned into 2-hour hourly bins (*e.g.*, 00:00–02:00 h, 02:00–04:00 h).

## RESULTS

### Structural variation of habitat types

PCA on pooled data between wet sclerophyll forest and rainforest reduced habitat variables into three major principal components that explained 42% of the total variation in vegetation structure. PC1, which explained 17% of the total variance, was correlated negatively with vines, palms, ferns, lateral cover and tree fall gaps, and positively with eucalypt emergents (Fig. 2; Table 2). PC2, which explained 15.2% of the total variance, was positively correlated with increasing distance to the nearest road, distance to the grassy forest edge, the number of treefall gaps, increasing slope and increasing rockiness (Fig. 2; Table 2). PC3, which explained 10% of total variance, was positively correlated with decay class of fallen timber, sub-canopy cover, and medium stem density, and negatively correlated with number of emergent eucalypts and density of ground cover (Table 2). Ecologically, PC1 therefore described a transition from rainforest (more palms, vines, ferns and lateral cover) to more sclerophyll-dominated forest (*e.g.*, more emergent eucalypts), PC2 described a trend from human disturbance (increasing distance to roads and grassy edges) to more natural disturbance (treefall gaps), and we interpreted PC3 to describe a trend in historical rainforest disturbance (perhaps from logging), where fallen timber in the plot was heavily decayed, medium stems density was high, and there was dense sub-canopy cover.

### Habitat correlates of *T. thetis* occurrence

Eight of the 48 cameras did not capture images of *T. thetis* on any of the 425 days of the study. Nineteen cameras photographed *T. thetis* on 1–10 days, seven cameras photographed *T. thetis* on 11–20 days, 12 cameras photographed *T. thetis* on 20–100 days and two cameras photographed *T. thetis* on more than 100 days (186 and 280 days). *T. thetis* occurred in both forest types, with no significant different between the number of independent events at cameras located in rainforest *versus* wet sclerophyll forest (Fig. 3). The occurrence of *T thetis* was significantly negatively correlated with PC2 (t = −2.985; *P* = 0.005) suggesting that *T. thetis* were more likely to occur at sites near roads and grassy edges, with few treefall gaps.

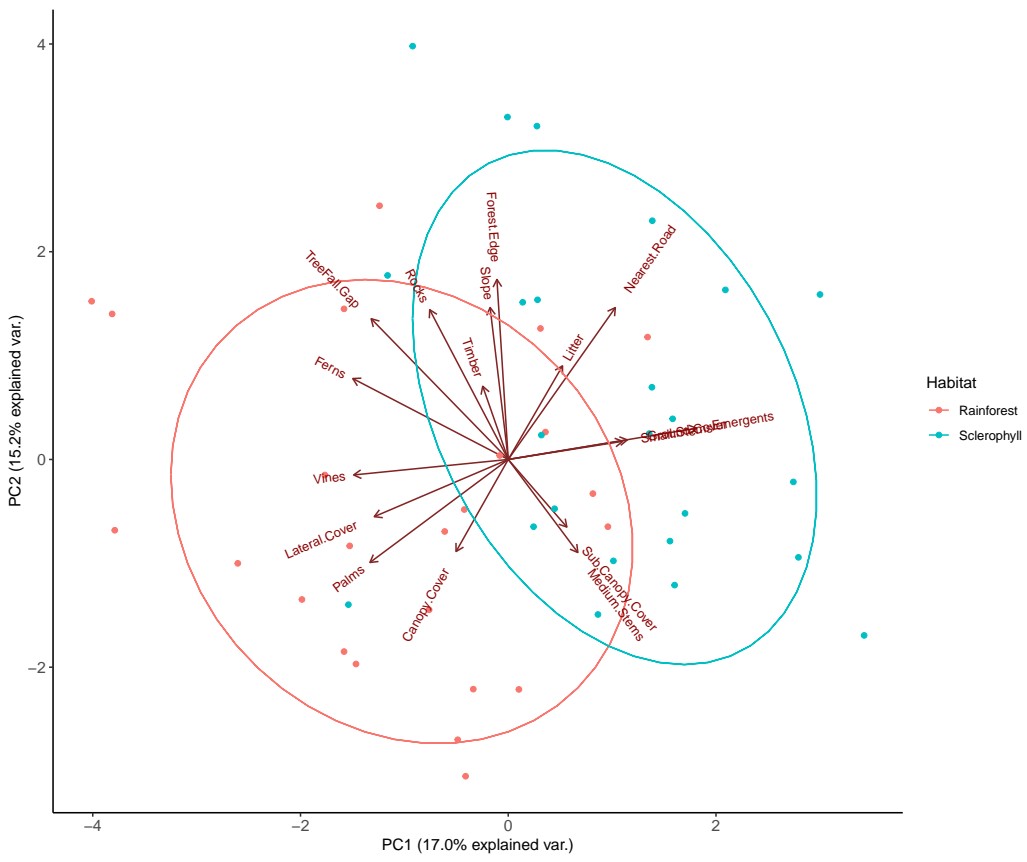

**Figure 2** **Principal component analysis (PCA) of habitat data measured at each camera trap location.** The components (PC1 and PC2) were extracted from an original dataset comprising 17 biotic and land-form variables that were chosen to reflect fine-scale habitat differences across the study area, with each point representing the score for a camera trap location. Ellipses represent 95% confidence level for a multivariate t-distribution.

## Habitat correlates of *T. stigmatica* occurrence

Fifteen of the 48 cameras did not capture images of *T. thetis* on any of the 425 days of the study. Cameras at a majority ($N = 26$) of the 48 sites photographed *T. stigmatica* on 1–10 days, three cameras photographed *T. stigmatica* on 11–20 days and four cameras photographed *T. stigmatica* on 20–100 days. The number of independent detections of *T. stigmatica* was not significantly different between cameras located in rainforest compared with those in wet sclerophyll forest ($p = 0.15$; Fig. 3). At a finer scale, independent detections of *T. stigmatica* were negatively correlated with PC1 (t = −2.643; $P = 0.01$), suggesting that *T. stigmatica* occupied forest containing elements more consistent with rainforest such as vines, palms and ferns that provided dense lateral cover. These sites also had few or no eucalypt emergents, but did contain natural treefall gaps.

## Temporal and spatial partitioning

Comparisons of the occurrence of *T. thetis* and *T. stigmatica* at individual cameras at which at least one of the species was detected showed a significant negative relationship between

Table 2 **Spearman rank correlation coefficients indicating the relationship of each of the 17 trap location attributes to the first three principal components (PC).** Negative (−) and positive (+) relationships are indicated for each, with asterisks (*) indicating level of significance.

| Measurement | PC1 | PC2 | PC3 |
|---|---|---|---|
| Leaf litter depth | | | |
| Vines | * (−) | | |
| Palms | * (−) | | |
| Ferns | * (−) | | |
| Rockiness of soil | | * (+) | |
| Fallen Timber | | | ** (+) |
| Ground Cover | | | * (−) |
| Lateral cover | * (−) | | |
| Canopy cover | | | |
| Sub-canopy cover | | | * (+) |
| Small stem density | | | |
| Medium stem density | | | ** (+) |
| Slope | | * (+) | |
| Tree fall gaps | * (−) | * (+) | |
| Eucalypt emergent | ** (+) | | * (−) |
| Distance to Nearest Road | | * (+) | |
| Distance to edge | | ** (+) | |

**Notes.**
*$P < 0.05$.
**$P < 0.01$.

the two pademelon species (t $= -2.708$, $df = 41$, $P = 0.01$); at cameras where *T. thetis* occurrence was relatively high, *T. stigmatica* occurrence was relatively low and vice versa (Fig. 4). Twenty-two cameras also returned low occurrence (less than the mean value) for both species, but only two had relatively high occurrences (greater than the mean value) of both species (Fig. 4).

Across the study site, *T. thetis* had higher rates of occurrence than *T. stigmatica* ($F_{1,80} = 13.06$ $p < 0.001$); while the two pademelon species had similar rates of occurrence in rainforest, in wet sclerophyll forest the occurrence rate of *T. thetis* was significantly greater than that of *T. stigmatica* ($p < 0.0001$; Fig. 3). In both forest types, *T. thetis* maintained an activity pattern where activity peaked at dusk (16:00–18:00 h) and dawn (06:00–08:00 h; Fig. 5). In rainforest, *T. stigmatica* was also most active at dawn, but with activity that extended into the late morning (08:00–12:00 h). However, in wet sclerophyll forest (where *T. thetis* had significantly greater occurrences than *T. stigmatica*; see Fig. 3), *T. stigmatica* was most active before dawn (04:00–06:00 h) and after dusk (08:00–24:00 h); times that corresponded with low *T. thetis* activity (Fig. 5).

## DISCUSSION

*T. thetis* and *T. stigmatica* demonstrated a degree of habitat partitioning at our study site, with occurrences of each species aligning with particular combinations of habitat variables. *T. thetis* occurred more commonly in locations close to grassy forest edges and forest tracks

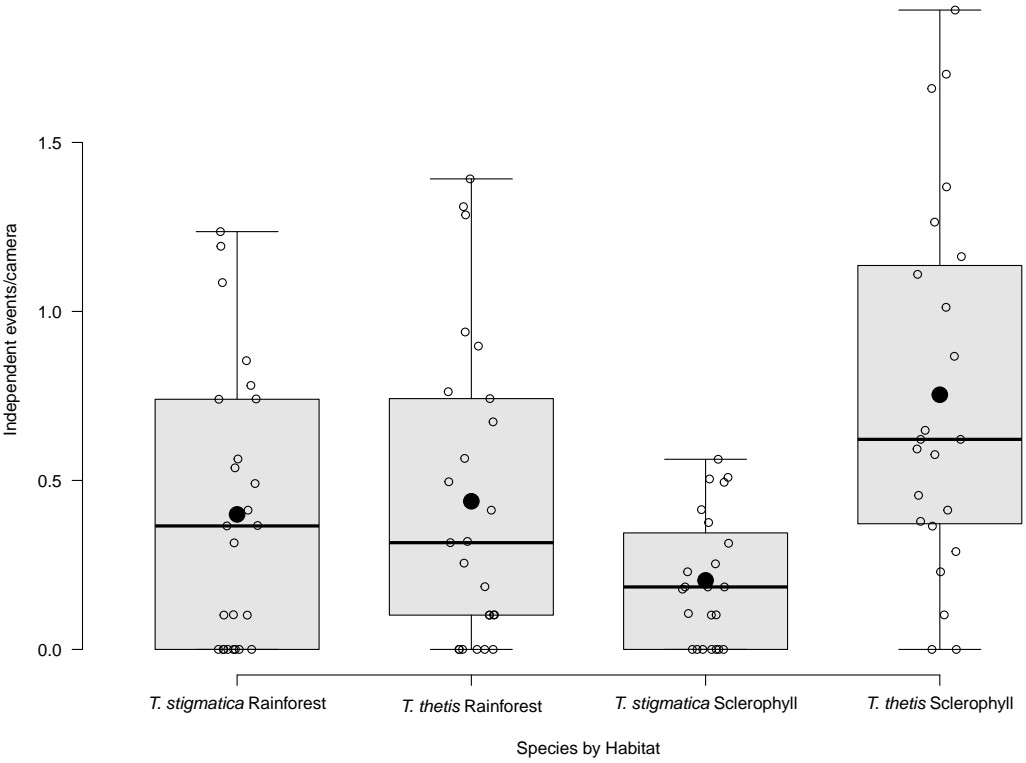

**Figure 3  Box and whisker plot of mean occurrence rate of red-necked pademelons (*Thylogale thetis*) and red-legged pademelons (*T. stigmatica*) at cameras located in rainforest or wet sclerophyll forest.** Open circles denote occurrence rate (days a species was encountered at a camera); closed circles show the mean. Boxes show the first to the third quartile, whiskers show the range, and black horizontal lines show the median.

rather than rainforest or wet sclerophyll forest. This result is easily interpretable in light of the occurrence of *T. thetis* at the abundant grassy resources found at forest edges (*Johnson, 1980*); by contrast, *T. stigmatica* at our study site has been shown by *Smith, Andrew & Vernes (2022)* to stay within the forest and not venture onto pasture. From our current study, we infer that *T. stigmatica* was more likely to occur in habitat containing rainforest elements (either rainforest, or sclerophyll forest with a rainforest understorey), with specific affiliation for vines, palms and ferns, and for sites with treefall gaps where vegetation at ground level was dense.

Home ranges typically scale with body size (*Harestad & Bunnell, 1979*); accordingly, small animals will perceive their environments at a fine scale and will be more sensitive to fine-scale vegetation structure and immediate landscape heterogeneity (*Stirnemann et al., 2015*). Sapling density, sub-canopy cover and medium stem density all provide sub-canopy cover for a ground-dwelling animal, and are examples of fine-scale measures (10s of metres) of vegetation heterogeneity. Because small-medium macropods shelter repeatedly at the same sites (*Jarman, 1991*), predator avoidance often depends on the ability to flee to a familiar location *via* known escape routes. Dingoes and wild dogs (*Canis familiaris*) are known to hunt pademelons (*Vernes, 2000*), and pademelons rely on dense vegetation
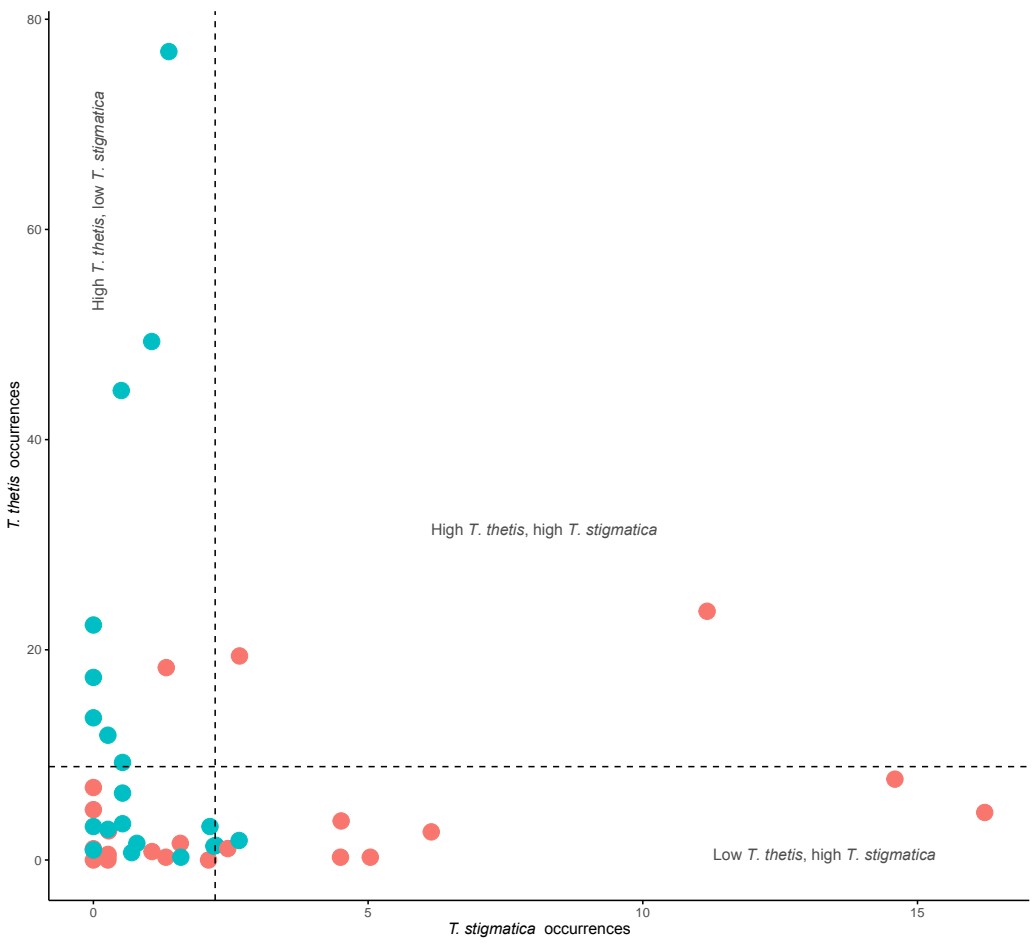

**Figure 4** **Comparison of the occurrence rate of red-necked pademelons (*T. thetis*) and red-legged pademelons (*T. stigmatica*) at each of the 48 camera traps (24 in rainforest and 24 in wet sclerophyll forest).** Blue circles are occurrence (total days a species was encountered at a camera) values for cameras located in wet sclerophyll forest; red circles are occurrence values from cameras located in rainforest. Dashed lines show the mean occurrence for each species, with text denoting zones of low (less than the mean) or high (greater than the mean) occurrence of each species.

to obscure them from such predators, particularly during the daylight hours (*Le Mar & Mcarthur, 2005*). Unlike other small macropods, *T. stigmatica* are active throughout much of the day and can move extensively throughout the forested parts of their range in daylight hours (*Vernes, Marsh & Winter, 1995*) in search of favoured rainforest browse (*Vernes, 1995*). Multilayered dense ground-layer rainforest vegetation would help to obscure *T. stigmatica* from predators, but would also offer them feeding opportunities for known food plants that include vines and ferns (*Vernes, 1994*). Treefall gaps would similarly offer browsing pademelons a diversity of pioneer species that thrive in high light conditions created by a treefall. *T. stigmatica* may frequent sites with these qualities more often due to the fitness benefits provided at a fine scale, such as cover from predators.

T. thetis Rainforest

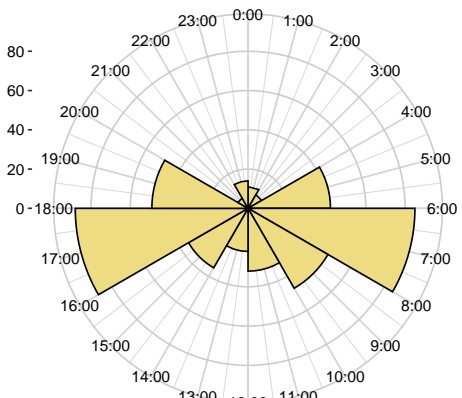

T. stigmatica Rainforest

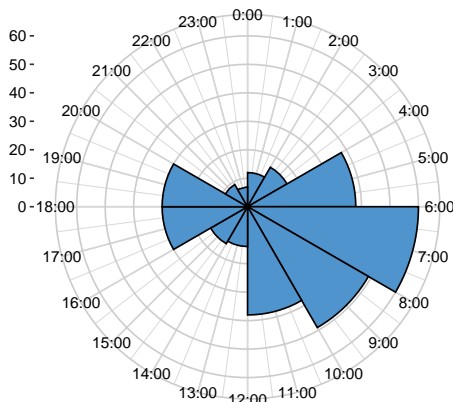

T. thetis Sclerophyll

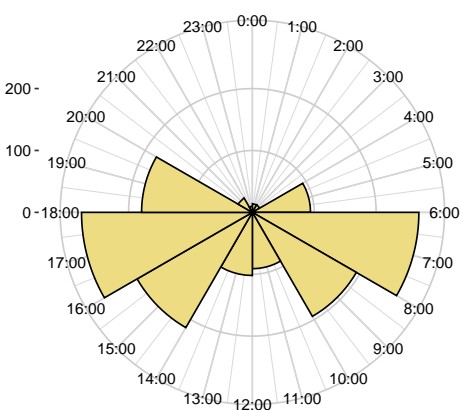

T. stigmatica Sclerophyll

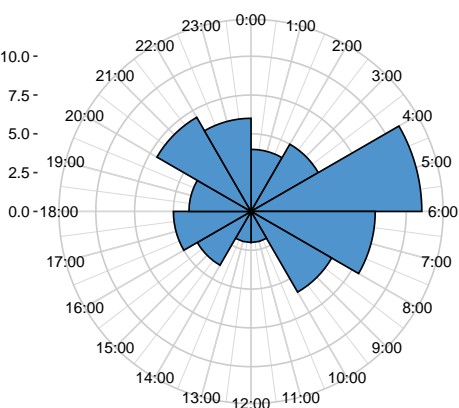

**Figure 5** **Polar plots of independent occurrences of red-necked pademelons (*T. thetis*) and red-legged pademelons (*T. stigmatica*) at cameras in rainforest or wet sclerophyll forest over the 24-hour cycle.** Occurrences are grouped into 2-hour bins for ease of comparisons. Independence for occurrences was defined as sets of photographs taken more than 30-mins apart. The 'count' on the *y*-axis provides a scale for the number of independent events comprising each graph.

Our findings detected more independent events for *T. thetis* than *T. stigmatica*, however, *T. thetis* did not occur in one forest type more than another. Rather, detections of *T. thetis* were associated with disturbance attributes like edges and roads, where grasses grow in the greatest abundance. This is consistent with previous findings that *T. thetis* exploit forest-pasture boundaries (*Jarman & Phillips, 1989*; *Johnson, 1980*; *Wahungu, Catterall & Olsen, 1999*) and our related study (*Smith, Andrew & Vernes, 2022*) that showed *T. thetis* was strongly crepuscular, with periods of heightened activity corresponding with foraging excursions to and from the forest edge and adjacent pasture.

Effects of fragmentation include both declines and increases in the abundance of some species due to alterations to the microclimate within the fragment (*Turton & Freiburger, 1997*). Our findings are consistent with earlier research that showed *T. stigmatica* does

not venture past the forest edge when in sympatry with *T. thetis* (*Jarman & Phillips, 1989*; *Smith, Andrew & Vernes, 2022*); by comparison, *T. thetis* makes regular foraging excursions beyond the edge into the adjacent pasture (*Smith, Andrew & Vernes, 2022*). Fragmentation increases the relative amount of edge to interior forest and significantly alters habitat at fragment edges (*Laurance et al., 2002*). Accordingly, *T. thetis* should occur at greater density than *T. stigmatica* at our study site because the site had a large grassy clearing at its centre, and roads and tracks that bisected the site, some of which offered grassy edges. Rather than associating with ubiquitous anthropogenic edges like roads and clearings, *T. stigmatica* may instead associate with tree-fall gaps that offer browsing opportunities for rainforest pioneer species away from grassy clearings.

When considering the conservation of species, understanding of habitat affiliation and niche utilisation is of obvious and paramount importance (*Vernes, 2003*). Competition avoidance in the form of altered use of space and temporal activity are likely employed by pademelons at our study site to facilitate co-occurrence. For example, *T. stigmatica* were detected on some cameras that were deployed very close to the forest edge, indicating that despite an apparent lesser association with disturbance and edge effects, *T. stigmatica* still utilised forest habitat all the way to the forest edge. However, cameras in the adjacent grassy clearing (see *Smith, Andrew & Vernes, 2022*) did not detect a single *T. stigmatica*, suggesting that they do not venture past the forest edge to graze pasture. Increased light penetration at forest edges can encourage an enriched understorey and a higher abundance of ground cover. This may at times attract *T. stigmatica* to edge-affected forest, however, competition with *T. thetis* probably excludes them from the adjacent pasture. In Australian vegetation communities, structural variation can govern the distribution of marsupials at various spatial scales (*Kanowski et al., 2001*). *T. thetis* appears to be spatio-temporally partitioning its habitat similarly to the way *T. stigmatica* does in the northern expanse of its distribution (*Vernes, Marsh & Winter, 1995*). Research on temporal activity of pademelons at our study site (*Smith, Andrew & Vernes, 2022*) also indicated some temporal partitioning between the species, suggesting that the two species are ecologically similar and subject to competitive interactions.

Prior to anthropogenic fragmentation of their habitat, pademelons (*Thylogale* spp.) are thought to have been edge-dwelling generalist species that exploited both rainforest browse and grasses in forested ecotones (*Vernes, 1995*; *Vernes, Marsh & Winter, 1995*). However, when sympatric with other forest-dwelling macropods, competition may force some species to narrow their niche breadth. In northern Australia, *T. stigmatica* occurs as the sole pademelon species, and there, habitat use and diet are very different from that seen in southern populations where *T. stigmatica* occurs in sympatry with *T. thetis*. These differences are also reflected in their cranial morphology; *Mitchell et al. (2018)* found that the southern subspecies of *T. stigmatica* (*Thylogale stigmatica wilcoxi*) had a broader cranium and a shorter and more robust muzzle—typical of browsing species, while the northern subspecies (*Thylogale stigmatica stigmatica*) possessed a more slender skull with a longer muzzle, a characteristic shared with *T. thetis* and that is commonly seen in grazing macropods. Direct competition for edge resources may have forced sympatric populations of *T. stigmatica* into a narrower niche and also driven their population density below

what might be achieved in the absence of competition; our related work (*Smith, Andrew & Vernes, 2022*; *Vernes, Elliott & Elliott, 2022*) indicated that *T. stigmatica* occur at lower population densities when in sympatry with *T. thetis* than when they occur in isolation from them. Thus, when constrained within a narrower, more specialised niche, population density of *T. stigmatica* may be reduced.

Species that have adapted to forest edges benefit from the fragmentation process whereas forest specialists have a higher tendency towards extinction, particularly where the home range of the species is not significantly smaller than the available fragment (*Harrington et al., 2001*). Disturbances like the removal of rainforest at landscape scales and the impacts of fires in sclerophyll forest have the capacity to negatively affect *T. thetis* into the future (*McHugh, Goldingay & Letnic, 2022*) and presumably, also *T. stigmatica*. However, *T. stigmatica* would likely be more affected than *T. thetis* if fragmentation and other anthropogenic disturbances were to increase, because *T. thetis* appears better adapted to the interface between forests and cleared land. Further research into habitat selection, diet, and niche specialisation in southern populations of *T. stigmatica* is therefore important for understanding their ecology and to help ensure their continued persistence. Because this study presents some spatial and temporal limitations in understanding pademelon co-occurrence, we also recommend further research at a landscape scale in a range of landscape settings to build upon the results we present. Nevertheless, our results indicate that protection of large tracts of rainforest where edge effects are minimised would clearly be advantageous for the conservation of *T. stigmatica* in the southern temperate parts of their range.

## CONCLUSIONS

The objectives of the current study were to investigate how structural attributes of two forest types, wet sclerophyll forest and rainforest, relate to the fine-scale occurrence of *T. thetis* and *T. stigmatica*. We found that *T. thetis* had similar occurrence in both forest types, but at a finer scale, was detected more at locations close to grassy forest edges and forest tracks where grasses were abundant. By comparison, *T. stigmatica* was more likely to be detected in rainforest habitat, with fine-scale affiliation for sites where vegetation at ground level was dense, including sites near tree-fall gaps. Our results therefore suggest that the threatened *T. stigmatica* requires tracts of undisturbed and unfragmented rainforest with fewer anthropogenic edges or incursions like roads.

## ACKNOWLEDGEMENTS

We are grateful to Rosemary Yates for her hospitality and access to her property 'Motherland' to enable this research to take place. We extend our thanks to Andrew Robertson, Peter Smith and Tim Henderson for their assistance in fieldwork and data collection. Thanks also to Rose Andrew who gave advice on some statistical analyses.

### Funding

This work was supported by HDR funds from the School of Environmental and Rural Science at the University of New England. The funders had no role in study design, data collection and analysis, decision to publish, or preparation of the manuscript.

### Grant Disclosures

The following grant information was disclosed by the authors:
School of Environmental and Rural Science at the University of New England.

### Competing Interests

Nigel R. Andrew is an Academic Editor for PeerJ.

### Author Contributions

- Lucy E.V. Smith conceived and designed the experiments, performed the experiments, analyzed the data, prepared figures and/or tables, authored or reviewed drafts of the article, and approved the final draft.
- Nigel R. Andrew analyzed the data, authored or reviewed drafts of the article, and approved the final draft.
- Karl Vernes conceived and designed the experiments, performed the experiments, analyzed the data, prepared figures and/or tables, authored or reviewed drafts of the article, and approved the final draft.

### Animal Ethics

The following information was supplied relating to ethical approvals (i.e., approving body and any reference numbers):

Ethics approval for this research was obtained from the University of New England Animal Ethics Committee (Approval No. AEC-1708).

### Field Study Permissions

The following information was supplied relating to field study approvals (i.e., approving body and any reference numbers):

Ethics approval for this research was obtained from the University of New England Animal Ethics Committee (Approval No. AEC-1708), and scientific licences for this research were issued by the New South Wales (NSW) Office of Environment and Heritage (Permit Nos. SL101721 and SL101837). While ethical approval specific to human subjects being caught incidentally on camera is not required in NSW, our standard procedure is to immediately delete any images of human subjects. However, as we set camera traps off-trail within a study site accessed via locked gates, no human subjects other than those directly associated with the project were captured.

### Data Availability

The raw data from camera traps locations are available in the Supplemental Files.

## Supplemental Information

Supplemental information for this article can be found online at http://dx.doi.org/10.7717/peerj.17383#supplemental-information.

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
