# Peer review of "Occurrence patterns of sympatric forest wallabies: assessing the influence of structural habitat attributes on the coexistence of Thylogale thetis and T. stigmatica"

_PeerJ, doi:10.7717/peerj.17383_

## Round 0.1 · original submission · Major Revisions

Overview
This study used camera traps to examine the habitat associations of two sympatric Thylogale species which differ in the amount of grazing included in their diet. Multiple habitat variables were measured around 24 camera traps in each of two forest types and combined in a principle components analysis. T. thetis occurrences were negatively correlated with PC2, which implied that they were more likely to be detected at sites with shorter distances to roads and forest edge, fewer treefall gaps, and flatter and less rocky terrain. T. stigmatica occurrences were negatively associated with PC1, which implied they were more likely to be detected at sites with more palms, ferns, lateral cover, and treefall gaps. The authors also reported that there were higher occurrence rates of T. stigmatica in rainforest than in wet sclerophyll but no difference between forest types for thetis. The findings are related to conservation strategies, especially for T. stigmata, a threatened species.

Both reviewers found this a generally sound manuscript but had a number of suggestions for improving the analysis as well as the clarity of presentation. Reviewer 1 asked whether it was possible to examine the influence of individual habitat variables. Reviewer 2 suggested that the study would be improved by a more rigorous analysis based on occurrence-based modelling and taking into account spatial autocorrelation and individual differences among traps. Although both reviewers suggested minor revisions, I have categorized the request as major revisions because changes may involve new analyses or at least a careful consideration of what is possible with your data set followed by an a well thought-out acknowledgement of any resulting limitations to the conclusions.

I have some additional concerns and suggestions which I summarize below. You may consider these as a third review, making changes when appropriate or explaining why changes are not necessary otherwise.

Editor’s Suggestions
General concerns

Many potential readers will be familiar with these species. It will help readers to keep track of which information applies to which species if you refer to them in the same order consistently throughout the manuscript. You frequently switch the order from the title, through the abstract, introduction, methods, results and discussion. I recognize that for some aspects such as geographical distribution, clarity and logic will determine which species you present first, but this will not be the case in general.

Clarification of study goals. I feel that there is a mismatch between the stated goals of the study and the approach used. This may require either clarification of how the methods fulfill the objectives or adjusting the objectives to match what was actually done. It is not clear to me how the objective of ‘investigating how the structural attributes of two forest types relate to the fine-scale occurrence of the two species’ can be achieved by an analysis that does not take forest type into account. Shouldn’t a reader expect at least a comparison of the two forest types for the variety of habitat variables measured? Wouldn’t an examination of ‘fine-scale habitat partitioning’ require some analysis of the positive or negative correlations of occurrence among camera traps?

Terminology of dependent variable. Reviewer 2 suggests that you avoid statements implying abundance because you (presumably) did not distinguish individuals. I agree with this but don’t agree that ‘activity’ is a better term because this is not what your measure reveals. I also do not agree that ‘preference’ (L204) is a valid term because that implies that individuals are making choices between alternative habitats, and your measure is some unknown combination of abundance and habitat choice. Perhaps ‘occurrence’ is the best term to use consistently, but you should check the literature to see what is the most appropriate and consistently used term for the type of measure you have.

Role of specific variables. I agree with the reviewers that attempting to determine which of the variables had the strongest associations with the occurrences would be valuable, especially for application to conservation, if it is possible.

Partitioning of the dependent variable. I wonder whether distinguishing occurrences that happened during the day, night or crepuscular period might yield any additional insights or at least could be reported in the descriptive section of your results. Also, did you consider the possibility of seasonal differences or breaking your study period into smaller time periods to provide some insights into seasonal variation and provide some replication within locations?

The Discussion is missing a clear examination of the strength of the conclusions. For example, should readers take into account the low average frequency of occurrences, the highly skewed distribution among cameras, limited number of camera locations, amount of variance explained by the PCs, ambiguities related to multiple habitat variables combined in PCs, or other aspects of the study that you would know much better than I? Does a PC that explains about 15-20% of the variance really indicate a ‘clear preference for rainforest’ (L253). I am not asking for a list of apologies but rather an indication to readers of how strongly they should take the conclusions. Are other studies needed to shore up some of the inferences?

The writing style is generally fine, but there are a few run-on sentences linked by ‘however’ and some misplaced commas. I have indicated these on the attached pdf.

The references are also quite well done but there are a few cases of missing volume numbers and authors’ names, capitalization of all words in journal article titles, and other minor errors. I highlighted those I noticed, but you should meticulously check all references for other errors.

Specific comments

Methods
L123. You state here that the species are similar in appearance but not in what way they differ. Since you were able to distinguish them from the camera trap images, I presume that there are visible differences. Mentioning relevant differences here will prepare readers for your explanation of how you distinguished them in the images and how confident you are in the reliability of that determination (also to be added).

L125. Is there any need for information about the phenology of the species, for example breeding season?

L134. It would be useful to briefly describe (with a reference) these two forest types and they ways in which they are similar and different for readers who are not familiar with them.

L147. This implies that the entire study area was off-limits to people except researchers. That seems relevant to the study area and should be clarified in the description of the study site above.

L164ff. This last part is not about camera trap placement but about the data extraction. There should be a whole paragraph on data extraction (probably requiring a change in the sub-heading), including how the species were distinguished, whether there was any ambiguity in this process, the definition of independent occurrences and the justification of the selected criteria for independence, and any other details needed for another researcher to be able to repeat the study.

L170ff. Is there any need for a mention of what month the measurements were made? Are any of the variables likely to change with season? If not, should that be stated?

L197ff. Although you have a goal of determining which forest type is favored by each species and do provide such a comparison, you report no statistical method for assessing this. Your method needs to account for the highly skewed distribution and large number of zeros.

L205. I was a bit surprised that your dependent variable is independent occurrences per trap. It is not clear whether you were able to set up all the camera traps on the same day. Most camera trap studies that I have read also have to deal with camera failures at different times. If these concerns apply to your study, I would have expected a measure such as occurrence per trap day (or perhaps per 100 trap days).

The reviewers both expressed some concerns about the true independence of your ‘independent occurrences’. Did you actually check in some way for temporal autocorrelation or was your choice of a 30-min interval an arbitrary choice? Did your video analysis suggest any explanation for the highly skewed distribution such as proximity to a regularly used sheltering or breeding site?

Results
L212. It seems awkward to refer to two components here and then list the first three. Do you need a justification to include three rather than two PCs?

L230. Considering the trend in means, would it be worth considering how much power you had to detect a difference? A lack of significant difference does not necessarily mean that they were the same as you seem to imply on L279 of the Discussion.

L242. I thought that PC2 related to distance from roads. Explain how the significant relationship to PC1 allows this conclusion.

L249ff. Several references in this paragraph to preference should be to occurrence. The discussion can address how preference can be inferred from this measure, but it is clearly an inference not a direct measure. Look at the published literature on habitat selection vs. habitat use that concern the same issue.

L252. Since you did not have camera traps out of the forest, I don’t understand how you can infer that stigmatica stayed within the forest from your study.

L257. I think that crypsis is usually used to refer to background matching, whereas hiding in dense cover is more likely to be called refuging. Please check the term.

L267. The contribution of this paragraph is unclear. It seems you made the point about refuges in the previous paragraph. What are you trying to convey here?

L292. You have not provided any analysis to show that thetis occurs at higher density than stigmatica. This requires a statistical comparison of occurrence rates between the species and a logical argument that occurrence differences reflect densities rather than other potential explanations.

Reviewer 1 ·

Basic reporting

The presented manuscript aimed to investigate habitat correlates for two sympatric pademelon species (Thylogale thetis and Thylogale stigmatica) in eastern Australia. The results of the study highlight that T. thetis moves between both rainforest and sclerophyll vegetation community types whereas T. stigmatica does not move beyond the canopy and does not graze in grassy vegetation types. Multivariate analyses revealed that T. thetis activity was correlated with disturbance factors such as roads and grassy forest edges whereas T. stigmatica activity was correlated with habitat elements within the rainforest. The authors conclude with some comments on conservation of rainforest habitat for T. stigmatica. Overall, the arguments made in the manuscript are well formed, it uses clear, appropriate and professional language throughout and the breadth of literature cited is comprehensive.

Experimental design

The multivariate analyses are appropriate for the data retrieved from camera trapping the two species, however, the grouping of covariates into broad categories waters down the detail to some degree. It would be interesting to see how the individual covariates correlate with each species activity. However, given the high degree of correlation between covariates, it is understood why the principal component analyses were undertaken and it is a suitable statistical approach for the detail required to answer the research objectives of the manuscript.

It is recommended that the authors refer to the dependent variable(s) in the analyses as activity indices. It is likely that individuals from each species could move between camera trap locations given their spacing and as such terms such as number of animals, abundance or occupancy/occurrence should be avoided.

Validity of the findings

The data support the notion that T. stigmatica have a somewhat contracted niche in the study area when compared to T. stigmatica populations in the north of its distribution. The study shows that when T. stigmatica is sympatric with T. thetis, it prefers to remain under the forest canopy to browse and does not move in to open grassy areas to graze where T. thetis is present. This is further supported by the principal component analyses that demonstrates that factors associated with rainforest were negatively correlated with T. thetis activity and factors associated with increasing sclerophyll components were negatively correlated with T. stigmatica activity.

Additional comments

General Comments
Line 29-31 it would be good if the authors could include some brief details on the statistical approach here

Line 33 – I’m note sure if more animals were detected unless you were able to identify individuals.
Perhaps rephrase to a higher number of detections of T. stigmatica were made in rainforest.

Line 34 was the metric used for these species’ occupancy or activity? Given the camera trap spacing one would think individuals could have moved between sites and therefore the term activity would be appropriate.

Line 35 replace ‘closeness’ with distance.

Line 69 Interesting, M. rufogriseus more likely to evade predators in open grassy habitat whilst P. billardierii more likely to evade predators in closed habitat. Also interesting that there are no Dingoes in Tasmania.

Line 73 remove ‘in order’

Line 76 – 78 Is there an example that you could use that demonstrates phenotypic changes resulting from competition between species? Fascinating.

Line 88 -89 Very interesting that T. stigmatica grazes on pasture in north Queensland. It’s as though when sympatric with T. thetis in southern areas it is forced to stay in the forest. Niche contraction?

Lines 99 – 101 perhaps productive tropical and sub-tropical rainforest allow T. stigmatica to occur in higher numbers when compared to warm temperate forest, considering its whole distribution.

Lines 103 – 106 are there any dietary studies that could be in cited here?

Lines 108 -112 this sentence could be shortened or divided in two

Line 114 change ‘forest’ to vegetation as one of the veg communities investigated was grassy area cleared of trees. However, when I look at figure 1 it appears that there were no camera traps in the open grassy area. Lines 151-152 state that there were 24 cameras in rainforest and 24 cameras in wet sclerophyll forest. Should the open grassy area be omitted from this section? Did the wet scherophyll forest have a grassy sub-formation? Given the emphasis on dietary preferences for the two species (browsing vs grazing) It would be worth mentioning.

Lines 164-165 was the vegetation maintained in the field of view of the camera as well during maintenance?

Lines 165-167 I see how you have derived the activity index now.

Lines 181 – 184 Was there any heterogeneity in vegetation cover within the sites? Was it possible to account for the variance/patchiness of vegetation cover within each site?

Lines 196 – Could the authors explain why the PCA method was used rather than using individual covariates?

Line227 add space between of and T. thetis and replace ‘occurrence’ with activity

Lines 232-234 is it possible to show a plot(s) outlining the influence of individual covariates on pademelon activity? or can they not be divided from the pooled PC groups?

Lines 241-243 as per comment above

Line 284 McHugh et al., 2022 found that red-necked pademelons had a negative relationship with % of cleared land within a 5km radius of camera trap sites which suggests that although this species will graze on forest edges at night, ample forest cover within this radius is a requirement. It would be interesting to know how far they are prepared to move from the forest edge.

Line 338 – Despite T. thetis being more common in the study area when compared to T. stigmatica, It might also be worth mentioning that disturbance such as fire frequency may have a negative effect on T. thetis (McHugh et al., 2022). High fire frequency may also change the mesic structure of Wet Sclerophyll forests in the Hyland area and may impact on T. stigmatica too. Might be worth mentioning given the recent wildfires across the Hyland and Chaelundi areas.

Line 467 Is it possible to include where the vegetation data were retrieved from?

Reviewer 2 ·

Basic reporting

1. Basic Reporting
• The manuscript is generally well written, with few sections of ambiguity. Some small areas need to be refined / grammar, spelling edited. e.g.,
o L48–49 grammar
o L49–51 The way this is written suggests that all cooccurring species must share the same resources. I would suggest editing to something among the lines of […. on which they specalise (resource partitioning); conversely, species that share resources may partition….]
o L63: “wield important” is unnecessary.
o L62–69, more detail on vegetation affecting spatiotemporal dynamics of cooccurring species would be important. I’m not sure exactly what to add, but this section feels a little too concise.
o L76–76, Might be useful to flesh that example out more (e.g., give details on the species, phenotypic traits). I think more specificity would help the argument being presented.
o L84–106, A very strong paragraph bringing together all the previously discussed.
o Just a suggestion: I think the first three paragraphs of the introduction could be re-arranged to strengthen the manuscripts arguments. For example, I would suggest (paragraph 1) further flesh-out the general introduction to resource partitioning in cooccurring species, with examples. I think the anthropogenic fragmentation section built in here is a little distracting and might be better suited as paragraph 4 (right before the study species section), but I’m not sure why this is important to the study, so some more elaboration would be useful as well.
 After reading the discussion it is clear fragmentation is important, so I would strongly recommend including more detail on fragmentation in the introduction.

• Figures
o Figure 1 title: use ‘New South Whales’ instead of the acronym here.
o L172 & Table 1: Not important but should fix: in text say concave densitometer and in table convex.
o Table 1: Add ‘%’ for Slope unit

Experimental design

2. Study design
• Field methods are well written and describe data collection very well.
o L171, please be specific of which overstorey characteristic is being referred.
• Statistical analyses need more explanation within the methods section, especially within lines 203–206.
o How were “independent events” at a camera trap defined? This is an incredibly ambiguous term in camera trap literature and how it is defined can have implications on results. How were species differentiated? Especially during low light conditions etc. I would strongly recommend defining independent events with a sufficiently long period of time between them (e.g., 1+ hours, Sollmann 2018) to remove potential bias from a single individual spending a long period of time in front of a camera. When glancing at the attached raw data, it seems that some cameras may be displaying this. Alternatively (below), this could be circumvented by using a binomial modelling approach.
• I understand this is a big ask, any may not alter the biological conclusions of the manuscript, but I hope I can effectively communicate my thoughts below. I think the analysis used in the manuscript could greatly benefit from some restructuring. This manuscript uses raw number of independent events from camera (i.e., photographic rates). Although I appreciate that this method is present in literature and has precedence, it is one of the least rigorous camera trap analytical methods (Sollman 2018, e.g., it is incredibly difficult to tell if number of photos were caused by extrinsic factors or have biological meaning). I would strongly suggest that the authors implement an occurrence-based modelling approach (Sollman 2018) to increase the statistical rigour and inference that can be made from the analysis. For example, using a Binary or Poisson distribution, the analysis could look at the binary presence-absence, or the count of times each species of wallaby appeared at a camera during a given time frame (e.g., daily, weekly, biweekly, monthly, seasonally—whatever makes the most biological sense for the study species)(Karanth et al. 2011; Mackenzie and Nichols 2004). This approach would also control for the potential temporal relatedness of species observations (e.g., L203–206, as a reader is it easy to presume many photos could have been the same individual walking back and fourth at a camera). This approach could also account for differential functioning of each camera trap if the authors implement a random intercept on a camera_id covariate (Gillies et al. 2006, Hebblewhite and Merrill 2008). Regardless of the approach used, the analysis should control for camera trap variation in some way. The degree of variation explained at camera traps may also merit an investigation of spatial autocorrelation given the cameras set up. I think that restructuring in this way would greatly improve the rigour of statistical analyses used in the manuscript.

Literature cited

Sollmann, R. 2018. A gentile introduction to camera-trap data analyses. African Journal of Ecology, 56(4): 740–749.
Karanth, KU., Gopalaswamay, AM., Kumar NS., Vaidyanathan, S., Nichols, JD., and Mackenzie, DI. 2011. Monitoring carnivore populations at the landscape scale: occupancy modelling of tigers from sign surveys. Journal of Applied Ecology, 48(4): 1048–1056.
Mackenzie, DI., and Nichols, JD. 2004. Occupancy as a surrogate for abundance estimation. Animal Biodiversity and Conservation, 27(1): 461–467.
Gillies, CS., Hebblewhite, M., Nielsen, SE., Krawchuk, MA., Aldridge, CL., Frair, JL. et al. 2006. Application of random effects to the study of resource selection by animals. Journal of Animal Ecology, 75: 887–898.
Hebblewhite, M., and Merrill, E. 2008. Modelling wildlife-human relationships for social species with mixed-effects resource selection models. Journal of Applied Ecology, 45: 834–844.

Validity of the findings

3. Validity of findings
• See below for specific examples. I would encourage including more of the summary statistics from the model output (including coefficient values and their respective significance within a plot or table).
o L230–232: This sentence should be citing Figure 2? also, we need to see the statistic the paper is citing here.
o L232, L241: Please clarify what is meant by “At a finer scale”. This analysis did not investigate spatial or temporal scale?
o L240–241: Should be citing Figure 2? Also need to see the statistic here.
o L248–249: “T. thetis showed no detectable preference for rainforest or wet sclerophyll forest..” I’m not sure where this result is being drawn from, and I think we would need to see model covariate values to validate this statement.
o L267–276: This section needs to be directly tied back to the manuscripts results. Here, it seems like an introductory paragraph brought in late and is distracting to the main story presented.

Additional comments

I very much enjoyed reading the manuscript and appreciate the work the authors are contributing on habitat partitioning on sympatric species with niche overlap. I would be happy to further review any potential future version of the manuscript. There are small areas of the manuscript that need some clean-up and elaboration, as well as some larger sections in the introduction and discussion that merit further explanation to fully articulate the narrative presented. Although the modelling framework presented here has precedence in literature, I would strongly suggest that the analytical methods used in the manuscript be slightly restructured to increase rigour and validity of findings. I have included a section below on how I best think this could be done. At a minimum, I believe that the analytical methods section be elaborated upon and with more context on justification on the methods implemented.

---

## Round 0.2 · accepted · Accept

The authors have taken a thoughtful and thorough approach to the comments from reviewers and editor and done a good job explaining their decisions. I find the manuscript substantially improved. I have attached a pdf with a few grammar and spelling comments to be corrected while processing the manuscript. Also, in Table 1 the criteria for the categories of fallen timber are not provided; rather the authors refer to a 1979 US Dept Agriculture handbook. If this will not be readily available to many readers, I think that the criteria should be made explicit in addition to providing the source of the concepts.